# The Use of Collagen-Based Materials in Bone Tissue Engineering

**DOI:** 10.3390/ijms24043744

**Published:** 2023-02-13

**Authors:** Lu Fan, Yanru Ren, Steffen Emmert, Ivica Vučković, Sanja Stojanovic, Stevo Najman, Reinhard Schnettler, Mike Barbeck, Katja Schenke-Layland, Xin Xiong

**Affiliations:** 1NMI Natural and Medical Sciences Institute at the University of Tübingen, Markwiesenstr. 55, 72770 Reutlingen, Germany; 2Institute of Biomedical Engineering, Department of Medical Technologies and Regenerative Medicine, Medical Faculty, Eberhard Karls University of Tübingen, Silcherstr. 7/1, 72076 Tübingen, Germany; 3Clinic and Policlinic for Dermatology and Venereology, University Medical Center Rostock, Strempelstraße 13, 18057 Rostock, Germany; 4Department of Maxillofacial Surgery, Clinic for Dental Medicine, 18000 Niš, Serbia; 5Department for Cell and Tissue Engineering, Scientific Research Center for Biomedicine, Faculty of Medicine, University of Niš, 18000 Niš, Serbia; 6Department of Biology and Human Genetics, Faculty of Medicine, University of Niš, 18000 Niš, Serbia; 7University Medical Centre, Justus Liebig University of Giessen, 35390 Giessen, Germany; 8BerlinAnalytix GmbH, Ullsteinstraße 108, 12109 Berlin, Germany

**Keywords:** collagen, bone tissue engineering, bone substitute materials, collagen modifications, composite bone scaffolds

## Abstract

Synthetic bone substitute materials (BSMs) are becoming the general trend, replacing autologous grafting for bone tissue engineering (BTE) in orthopedic research and clinical practice. As the main component of bone matrix, collagen type I has played a critical role in the construction of ideal synthetic BSMs for decades. Significant strides have been made in the field of collagen research, including the exploration of various collagen types, structures, and sources, the optimization of preparation techniques, modification technologies, and the manufacture of various collagen-based materials. However, the poor mechanical properties, fast degradation, and lack of osteoconductive activity of collagen-based materials caused inefficient bone replacement and limited their translation into clinical reality. In the area of BTE, so far, attempts have focused on the preparation of collagen-based biomimetic BSMs, along with other inorganic materials and bioactive substances. By reviewing the approved products on the market, this manuscript updates the latest applications of collagen-based materials in bone regeneration and highlights the potential for further development in the field of BTE over the next ten years.

## 1. Introduction

Increasing incidence of bone defects due to trauma, disease, or tumor resection has given rise to a growing need for bone grafts [1]. Currently, for the surgical treatment of bone defects, autologous bone grafts still represent the “gold standard” for bone repair [2]. However, the clinical applications of autografts are restricted by the limited donor tissue availability and potential for severe postoperative complications [3]. To overcome these limitations, intensive investigations in bone tissue engineering (BTE) and material science have been carried out to produce ideal bone substitute materials (BSMs) as an alternative to autografts [4,5,6]. So far, various BSMs have been developed, and some have even entered clinical use, including metals and alloys (e.g., stainless steel, titanium alloy, and magnesium alloy) [7], minerals (e.g., hydroxyapatite and tricalcium phosphate among, other calcium phosphate compounds) [8], synthetic polymers (e.g., polyurethan) [9], naturally derived polymers (e.g., collagens and gelatin) [5], as well as a significant number of composites of these materials [10,11]. However, there is much room for improvement in order for these biomaterials to become real autograft substitutes and achieve successful bone regeneration. Incomplete or defective regeneration can lead to loss of tissue mass or replacement with fibrotic scars, which are associated with impaired functional recovery [12]. Angiogenesis is another big challenge for synthetic BSMs, and even BTE constructs, as an insufficient blood supply can lead to necrosis and ultimately, to failure of the bone replacement [13]. Therefore, although clinical research in synthetic bone grafting has been conducted for more than a century, only a few of these methods are available for long-term clinical use (Table 1 and Table 2). As the main component of bone matrix, collagen (type I) and its engineered forms plays an increasing important role as a component of bone substitute materials and in BTE because of its excellent biocompatibility, cell adhesion, and osteoconductivity [4,14,15,16]. However, its shortcomings including poor mechanical properties, high degradability, and lack of osteoinductivity currently limited to date routine clinical applications [4,14,17,18]. For this, many attempts have been made to improve collagen-based implants in bone tissue repair and engineering. Hundreds of review articles have been published in the field summarizing the findings regarding collagens and collagen derivatives for BTE applications, and most of these articles focus on new trends in scientific understanding and research; however, the key issues of clinical usability and technology transferability are rarely addressed. In this short review, based on a summary of information on currently marketed products with FDA/EU market authorization, an overview of collagen resources, preparation methods, advancements and shortcomings in modification techniques, the use of collagens for clinical bone grafting, and the development of more functionalized collagen-based composites for BTE are presented. For more than 200 years, collagens have been studied as the most abundant naturally derived biomaterial, and many techniques and methods have been extensively developed and adapted during this time. However, as techniques improve and new knowledge emerges, some theories must be continuously revised and updated, albeit subtly, such as in the exploration of many chemical modifications and mechanisms of cross-linking reactions, or in the involvement of collagen in the regulation of different (patho)-physiological processes, especially clinical phenomena such as immune responses.

## 2. Collagen in Native Bone Tissues

Collagen (type I) is a major structural component of mammalian bone, constituting 90% of the organic components of the bone extracellular matrix (ECM) [19]. Thus, it has been stated that the ideal synthetic bone grafts should mimic the ECM of autologous bones as much as possible, since the ECM found in natural tissues supports cell attachment, proliferation, and differentiation [20]. Scaffolds should consist of appropriate ECM-like biochemistry and nano/micro-scale surface topographies in order to formulate favorable binding sites to actively regulate and control cell and tissue behavior [17,21]. Therefore, understanding the synthesis, structure, and distribution of collagen in bone tissue is critical for bone regeneration [17]. The process of collagen synthesis in bone tissue occurs mainly by fibroblasts and osteoblasts [22]. Within the cells, polypeptides formed in the endoplasmic reticulum are the building blocks of collagen, called alpha chains [23,24,25,26]. After translation and posttranslational modifications, three alpha chains are linked by disulfide bridges [27], then twisted into a triple helical procollagen [28]. The ability to form a triple helix is the most important property of collagen, which is mostly based on the repetitive Gly-X-Y triplet of alpha chains [22]. Glycine, with the smallest volume, is located at the inside of the triple helix. X is usually a proline, but could also be other amino acids; and Y is often hydroxyproline. The alpha chains are twisted in a left-handed way, while the triple helixes are folded in a right-handed helix because of the presence of hydrogen bonds. The formed triple helical procollagen is excreted into ECM by transport vesicles. In the ECM, the globular N- and C-terminal propeptides are removed from procollagen by the N- and C- proteinases, which is the key step in the formation of mature collagen [25]. The helix dimension is on the order of 1.25 nm wide and 300 nm long [29], with a mass of ~285 kDa. The removal of the N- and C- terminal propeptides would trigger the self-assembly of supramolecular structures (collagen microfibrils, fibrils, and fibers). Finally, the covalent crosslinking and non-covalent bonds stabilize these supramolecular structures [30].

In bone, groups of five triple helical collagen molecules form microfibrils. The microfibrils are spontaneously organized into fibrils and fibers that can be up to 1 cm in length and 1 mm in diameter and with the characteristic 67 nm banding feature, called the D-period [31,32]. The hydroxyapatite (HA) nanocrystals are deposited by osteoblasts on the collagen fibrils, constituting the inorganic and organic phases of the bone matrix. The bone matrix is thus hierarchically structured, consisting of a 65% mineral phase, HA, a 35% organic phase (~90% type I collagen, 5% noncollagenous proteins, and 2% lipids by weight), and a residual amount of water (Figure 1). All these scales play an important role in the remarkable mechanical properties of bone. HA provides the rigidity, and collagen fibers improve the toughness of bone. Thus, to better mimic the native bone tissue, collagens used in synthetic bone materials should maintain their natural structure as much as possible. This imposes high requirements on collagen sources and extraction techniques.

## 3. Collagen Sources and Extraction Techniques

Collagen can be extracted from most animal tissues. Rat-tail tendon collagen is widely used in basic research due to its high purity and homogeneity, and its simple extraction process [33]. For the medical devices, bovine skin and tendons, as well as porcine skin, are the most common collagen sources, with collagen type I as the predominant fraction. They are highly homologous to human collagen, with almost no immune restriction; therefore, they are often used for hernia repair and wound healing [34]. Collagens isolated from other animal sources, such as the sternal cartilage from domestic birds, are also available, but they are not in wide clinical use [35]. Recently, collagens from the ocean (such as from sea sponges and jellyfish) have attracted interest because of their huge potential reserves and batch-to-batch consistency [36]. However, their clinical validation and use still require long-term research and development.

Collagen isolation can essentially be obtained by chemical extraction and enzymatic processing [37]. Chemical extraction is more common for industrial scale production, including neutral saline solutions (e.g., sodium chloride (NaCl), Tris-HCl, phosphates) [38,39] and acidic solutions (e.g., acetic acid, citric acid, lactic acid, and hydrochloric acid) [40,41]. Acidic extractions with organic acid solutions, especially acetic acid, are able to solubilize non-crosslinked collagen and also break some of the inter-strand cross-links in collagen, achieving a higher productivity of collagen molecules [37]. Enzymatic processing with selected enzymes (e.g., pepsin, papain, and collagenase) shows some advantages, such as specificity, control of dissolution, mild reaction conditions, and less waste generation; for these reasons it is more widely used, but there are also medical concerns about enzyme inactivation and purity [42,43,44]. Nevertheless, extraction methods may favorably and unfavorably affect the final characteristics of collagen, including thermal stability, molar mass, water retaining ability, and gel-forming capacity [37]. Although marine collagens and collagen peptides are considered potential alternatives to mammalian collagens, their low thermal and mechanical stability still need to be improved by chemical cross-linking and other processes. Moreover, the chemical and structural differences compared to mammalian collagens should not be underestimated for potential medical applications.

To avoid problems related to transmission of disease and cumbersome extraction processes induced by animal extracted collagens, the production of recombinant human collagens (rhCol) appears to be an alternative option [45]. However, the complex post-translational modifications are critical for the production of rhCol, which cannot be fulfilled in unicellular organisms [33]. In most cases, the produced rhCol showed insufficient enzymatic or thermal stability, suggesting incomplete triple helix and fibril/fiber formation [46]. Moreover, the quality control, including batch-to-batch consistency, as well as the genetic construction of the bioengineered host strain, are still under development [47], and so far, the yield is too low for industrial applications [46]. Altogether, even though the sources of collagen are more diverse and the extraction and engineering techniques are becoming more advanced, the mechanical strength and biological stability of the obtained collagen products are still not comparable to those of native collagen fibers. Further in-depth explorations of collagen sources and extraction techniques are still necessary.

## 4. Exogenous Collagen Modifications

To achieve effective bone repair, collagen-based bone materials should match the mechanical properties of bone in the early stage of bone regeneration, and then gradually degrade and be replaced by regenerated bone tissue [48]. For this purpose, collagen must be modified (e.g., cross-linked or blended) for specific medical application and clinical indication. Thus, it is often necessary to introduce chemical, physical, or biological methodologies resulting in natural exogenous crosslinks (as shown in Table 3) into the inter-/intra-molecular structure to tune the mechanical properties, to prevent denaturation at 37 °C, and to control the degradation rate [33,49,50]. The fundamental principle of exogenous collagen cross-linking is the formation of covalent bonds between collagen molecules using chemical or natural reagents, typically linking the free amine or carboxyl groups of collagen (Figure 2). However, only a few devices are currently on the market because it has been shown that most of the collagen crosslinking technologies lead to undesirable inflammatory tissue responses. Therefore, research in this field is still highly topical and is constantly being pursued by industry and universities alike.

### 4.1. Chemical Methods

Chemical cross-linking is well known, mainly from other industries, such as leather manufacturing, but this technique has been transferred to the medical device industry for decades due to its high crosslinking efficiency. The widely used chemical cross-linking agents are aldehydes (e.g., glutaraldehyde, GA [51,53]), isocyanates (e.g., hexamethylene diisocyanate, HMDI [53,54]), carbodiimides (e.g., 1-ethyl-3(3-dimethylaminoproopylcarbodiimide and N-hydroxysuccinimide, EDC-NHS [55,57]), genipin [52], and citric acid [61,73], with varying degrees of efficiency. For many years, GA processing has been the “gold standard” method for collagen crosslinking, since it could react with all the available free amine groups of collagen molecules, forming a very strongly crosslinked network [51]. However, the aldehyde groups of GA are cytotoxic, which would cause severe inflammation in the body, in addition to biohazard problems [74]. This limits the further applications of GA-crosslinking in medical devices. A water soluble carbodiimide system, EDC-NHS, was then expected to be a safe alternative crosslinking method for biomaterials, as well as for different clinical applications [75,76]. However, the biomechanical properties and degradation profiles of such materials are not as desirable as those for the GA-crosslinked examples [77]. In recent years, green chemical crosslinkers, such as genipin, citric acid, and tannic acid, have attracted increasing attention from researchers due to their biodegradability and low cytotoxicity [58,61,73,78]. However, until now, most of them have remained at the experimental stage, and there is no economic justification for industrial production [51]. In addition, collagen co-polymerization is also an effective method for collagen modification, contributing to fibril organization and improving the mechanical properties of collagen [79,80]. A further sugar-mediated (e.g., ribose) crosslinking method, called glycation, is also in process [81,82]. The glycation crosslinked collagens showed a > 5 times higher resistance to enzymatic degradation [83]. However, the inflammatory tissue response and cellular reaction on crosslinked collagens are still not fully investigated [83]. Considering the increasing requirements for market authorization, authorities need more detailed knowledge and in-depth quantitative analyses to demonstrate the safety, efficacy, and process stability of these products. The qualitative and quantitative determination of the cross-linking mechanism, the sites of cross-linking, the homogeneity, and the reproducibility of the reaction still fail to meet the regulatory requirements. New knowledge is needed, and new techniques are under development.

### 4.2. Physical Methods

Physical methods, such as dehydrothermal, lyophilization, UV irradiation, with or without photoactive agents (e.g., rose Bengal and riboflavin), could improve the physical and mechanical properties with negligible toxicity and less concern about cross-linker residues compared to chemical methods [18]. Lyophilization is the main established physical method for stabilizing and structuring collagen [84,85,86]. Based on the freeze-drying process and the regulation of the thermodynamic rate, molecules and small fibers can be aligned together due to the formation of ice crystals, which is related to the solvent used [87]. The structure of collagen can alter many physical properties of the scaffold, including pore size, pore distribution, arrangement of open pore structures, wettability, and infiltration. Further, these properties can regulate cell signaling, cell adhesion, and other biological activities [88]. This approach is one of the most approved techniques for the industrial production of collagen-based devices such as wound dressings and cartilage fillers. Riboflavin and UV induced corneal collagen crosslinking (R-UV-CXL) has been well-recognized as an effective treatment for progressive keratoconus for many years [89]. Recently, R-UV-CXL has gradually entered the research field of medical devices and BTE. For example, Bapat et al. reported that 3D-printed collagen scaffolds impregnated with quaternary ammonium silane and crosslinked with riboflavin and UV produced a promising scaffold with antimicrobial potency and structural stability [90]. R-UV-CXL has also been studied in the fields of wound healing [91], skin repair [92], cardiac valves [93], and vascular regeneration [94]. A study by Vasilikos et al. on the R-UV-CXL treatment of intervertebral disc (IVD) matrices suggested that this treatment may be a promising tool to reinforce the IVD matrix [95]. In addition, some other photocrosslinking techniques have also been proposed. For example, biocompatible hydrogels crosslinked with lumichrome and blue light, proposed by Grønlien et al., exhibited increased elasticity, water absorption properties, and water retention capacity compared to R-UV-CXL [91] (Figure 3). However, the adverse structural changes due to over-treatment and the inconsistent degree of crosslinking caused by the insufficient irradiation on the larger or three-dimensional scaffolds should not be ignored in physical collagen modification.

### 4.3. Biological Methods

Tissue-type and microbial transglutaminase (TG) have been utilized to stabilize collagen- and gelatin-based materials, mimicking the enzymatic in vivo collagen crosslinking pathway [33]. The TG mediated crosslinking served to increase in denaturation temperature, mechanical strength, and biological resistance, while showing no cytotoxicity to cells [47,96,97,98]. Recently, a study on recombinant human transglutaminase 4 (rhTG-4) supplementation in a hyaluronic acid/collagen/fibrinogen (HA/COL/FG) composite gel reported that its addition significantly upregulated the aggrecan and type II collagen mRNA of stem cells, increasing the hardness of the ECM [99]. However, it should be noted that TG-based biochemical collagen modifications, despite their excellent biocompatibility compared to chemical crosslinking methodologies, showed a much lower crosslinking efficiency, often lower than that of the mildest chemical approaches [33]. Moreover, the high price of the TG-based approaches also limits their industrial application.

Therefore, the quest for an optimal collagen crosslinker continues. Further efforts to modify collagen are still under development, e.g., the coupling of functional peptides, proteins [100,101,102], different ECM-components or derivates (e.g., glycosaminoglycans and Decorin) [103,104], polymers (e.g., PLLA and PLGA) [105], and other composite materials such as bioglass and PCL composites [106].

## 5. Applications of Collagen in Bone Tissue Regeneration and Engineering

The earliest reports of the use of collagen in the biomedical field date back to the 19th century [107]. Currently, collagens directly extracted from animal tissues, or produced as recombinant proteins, with or without further modifications according to crosslinking, polymerization, or fibrillization, have been widely applied in vitro as standardized 3D materials to investigate the influence of microstructural and mechanical features on cell behaviors, such as cell attachment, cell contraction, cell motility, and related gene expressions [108,109]. Collagen-based materials in various forms, including membranes, sponges or matrices, hydrogel, and composite scaffolds, are also widely used in vivo to support bone tissue regeneration in different clinical applications [18,110] (Figure 4).

### 5.1. Collagen Membranes

Various kinds of membrane materials are applied in the clinical practice, such as in the field of dentistry, to achieve guided bone regeneration (GBR) [111] (Figure 5).

Expanded polytetrafluoroethylene (ePTFE; Teflon) presented the first success for controlling the migration of soft tissue, and especially epithelial cells, to achieve GBR [112]. However, with the wide use of ePTFE, this kind of non-resorbable membrane shows a high rate of wound dehiscence and infections, and often needs to be removed by a second operation, which causes a secondary damage to the wound [113,114]. One promising alternative to overcome these concerns is the application of resorbable collagen membranes, which offer many advantages, such as a single-step surgical procedure, improved soft tissue healing, the incorporation of the membranes by the host tissues, and a quick resorption in case of exposure [115]. However, the disadvantages of collagen membranes are related to their unfavorable mechanical properties and their unpredictable resorption rates, even with the use of native dermis-derived materials [116] (Figure 6).

Thus, different approaches, such as the choice of other tissue sources and manufacturing methods, have been tested to overcome these issues [81,83]. Interestingly, cross-linking based on ribose has been shown to be a favorable alternative for the production of collagen-based GBR membranes [82]. Other methods to overcome these issues are the use of new manufacturing approaches, such as electrophoretic deposition, electrospinning, and 3D printing, to improve the mechanical and biological properties of collagen membranes [117,118]. Recently, another potential material alternative, based on a resorbable magnesium grid for volume stability, combined with a collagen membrane or a pure magnesium membrane, have been developed for GBR procedures [119,120].

### 5.2. Collagen Sponges or Matrices

Collagen sponge is one of the most useful biomaterials owing to its excellent function and properties, as well as its easy processing, sterilization, and preservation [121]. Collagen sponges are generally formed using a freeze-drying process (also known as ice-crystal templating, lyophilization, or ice-segregation-induced self-assembly) (Figure 7) [122,123]. By altering the freezing conditions, such as freezing temperature, time, and molds, the pore size and shape of collagen sponge could be tailored [124]. For the optimal bioactivity of tissue regeneration, the pores should be large enough to permit cell migration and nutrient diffusion, and small enough to promote cell attachment [125,126].

FDA approved collagen sponge-based devices are mainly used as absorbable hemostatics in different clinical conditions (Figure 8). For BTE, collagen sponges are mainly applied as basic scaffolds for carrying bioactive substances, including growth factors, cells, drugs, etc. Recently, a collagen sponge, combined with BMP-2 and a bridge protein, showed the improved safety of BMP-2 and effects on bone regeneration for spinal fusion [127]. A gene delivery system encoding fibroblast growth factor (FGF-2) and BMP-2 embedded in collagen sponge showed optimal bone regeneration effects [128]. Atelocollagen molecules self-assembled into collagen fibrils, then processed with freeze-drying and crosslinking to build up a collagen sponge scaffold, which showed improved osteogenic differentiation and bone formation in vitro and in vivo (Figure 9) [129]. In addition, collagen sponges play an important role for in vitro models. To mimic the complex hierarchical environment of the native ECM, experimental design tools and optimized freeze-casting systems present exciting opportunities for the tailored architectural design of ice-templated collagen sponge scaffolds [130,131].

### 5.3. Collagen Hydrogels

Hydrogels are water-swellable polymer materials with a 3D network structure formed through crosslinking reactions [132]. Naturally derived collagen hydrogels present more satisfying biocompatibility and biological activity compared to other synthetic polymers or hybrid polymers [133]. Similarly, they exhibit low stiffness and rapid degradation, which hamper their application in animal models and clinical tests [134]. To overcome these drawbacks, one approach is based on the self-assemble ability of collagen hydrogels, as they polymerize into a fibrillar structure at physiological pH and ionic strength and temperature following an entropy-driven process [135,136]. Another efficient strategy is the production of composite hydrogels formed by the combination of collagen hydrogels and synthetic components, including functionalized polymers and organic/inorganic nanoparticles and ions [132]. The hydrogel collagen nanocomposite, in combination with strontium and seeded mesenchymal stem cells, showed the highest radiographical and histological scores compared with only collagen hydrogel and other control groups in full-thickness bone defect regeneration in the rabbit model [137]. The application of nanomaterials to improve the mechanical properties of hydrogel showed enhanced stem cell adhesion and provide valuable guidance for the design of hydrogel-based materials [138]. Currently, a new generation of composite collagen hydrogels is widely applied as an injectable hydrogel scaffold for in situ bone tissue repair, flexible drug delivery systems (nanogels or microgels), and implanted bone tissue scaffolds using 3D-printing, electrospinning, or other techniques [132].

### 5.4. Collagen-Based Composite Materials

#### 5.4.1. Collagen Integrated with Organic or Inorganic Materials

In FDA and EU approved collagen-based composite material (Table 1 and Table 2), collagen bone void fillers (CBVFs) are one of the protagonists, and they are used for the backfilling and structure stabilization of bone voids in orthopedic surgeries. CBVFs are commonly composed of inorganic materials, such as calcium sulfate or calcium phosphate augmented with hydroxyapatite, and a native tissue-derived collagen suspension or collagen hydrogels, providing improved biocompatibility, resorbability, and filling ability [139]. In recent decades, CBVFs and other collagen-based scaffolds have been introduced into various materials, such as hyaluronic acid, alginate, glycosaminoglycans (GAGs), silk fibroin (SF), metals, bioactive glasses, and some novel carbon-based materials [76,78,140,141,142,143]. For example, a mechanically robust, injectable, and thermoresponsive CBVF was formed with carboxylated single wall carbon nanotubes (COOH-SWCNTs), chitosan, and collagen, as reported by Kaur et al. [144]. They found that the material showed improved mechanical properties and bioactivity with the addition of COOH-SWCNTs [144].

#### 5.4.2. Collagen-Based Composite Materials Loaded with Growth Factors, Cells, or Drugs

Although collagen type I fibers contain multiple biological cues that could directly interact with cells to modulate their adhesion, proliferation, and differentiation, largely improving the biocompatibility and osteoconductivity of bone substitute materials, collagen alone is not osteoinductive [14]. Thus, to achieve effective bone replacement and sufficient tissue vascularization, collagen-based materials can be modified to include other bioactive substances, such as growth factors (e.g., BMPs, VEGFA, etc.), cells (e.g., BMSCs, osteoblasts, osteoclasts, HUVECs, etc.), and drugs (e.g., antibiotics). The first collagen bone graft loaded with recombinant human bone morphogenetic protein-2 (rhBMP-2) was approved by the FDA in 2002 [145]. However, after an initially promising start, concerns regarding safety and cost-effectiveness of BMPs have been raised [146]. A solution may be the optimization of the delivery system of BMPs to decrease the need for high doses of BMPs and to prevent their applications [147]. In addition to increasing the loading efficiency by these modifications, other attempts have been made by inhibiting the BMP antagonists and the combination of multiple growth factors [148]. In addition, collagen materials could be integrated with drugs for antibacterial, osteogenesis, and angiogenesis activity. Nabavi et al. reported that collagen hydrogel loaded with tacrolimus, which can enhance osteogenic differentiation by activating BMP receptors, showed improved bone formation compared with that of other non-tacrolimus groups [133]. Because of the interactions between the RGD (arginine-glycine-aspartate) sequence and stem cell integrin receptors, collagen-based materials can increase the adhesion, proliferation, and differentiation of stem cells, providing an ideal platform for cell delivery [133]. Recent studies have reported some rational designs of materials by surface modification and stiffness adjustment with nanomaterials to tune the tether mobility and anisotropic nanoscale presentation of RGD [138,149,150]. A collagen scaffold carried with dental pulp stem cells (DPSCs) showed a higher amount of calcification in the reconstructed defects [151]. However, in another study, in vivo micro-CT analysis confirmed that the acellular scaffolds generated larger volumes of bone than the DPSCs seeded scaffolds [152]. Some other attempts at creating collagen-based composites are also focusing on the integration of collagen with peptide delivery and gene therapy [153,154,155,156]. These studies remain controversial and have yet to be explored. Another promising direction is the synergistic effects of multiple bioactive cations on vascularization and bone defect repair, which is a much safer and simpler option [157].

## 6. Conclusions and Perspectives

Significant strides have been made in the field of collagen (type I) for bone tissue engineering. Advances in exploring various collagen sources, as well as extraction and purification processes, have made collagen preparation available with maximum native structure and minimum immunogenicity/antigenicity. Advances in collagen modifications have offered optimized collagen mechanical properties and biological resistance. For a wide variety of applications in bone grafting, collagens are often modified or combined with other materials to construct various bone substitute materials, such as collagen sponges, hydrogels, nanofibers/microfibers, or nanoparticles/microspheres. Collagen-based composites with bioceramic materials (e.g., HA, TCP, and BGs, etc.), carried with growth factors, peptides, cells, drugs, and genes, have made the multifunctionalized BSMs available with improved osteoconductivity, osteoinductivity, osteointegration, osteogenesis, and vascularization. Although many of them have shown great effects on bone regeneration in in vitro and in vivo studies, only a few of them are currently FDA- or EU- approved for clinical applications. These composites still face challenges regarding complete mechanical properties, biological stability and activity, immune response, regional vascularization, and other safety issues for human bone regeneration, which hinder their translations into clinical devices. However, in addition to the continuous development of new tools and technologies for material manufacturing, such as 3D printing and electrospinning, researchers are directing attention to patient-specific collagen-based materials. Rather than relying on the material alone, an integration of functional material manufacture and the innate regeneration potential of patients sheds new light on potential clinical translation within the next decade.

## Figures and Tables

**Figure 1 ijms-24-03744-f001:**
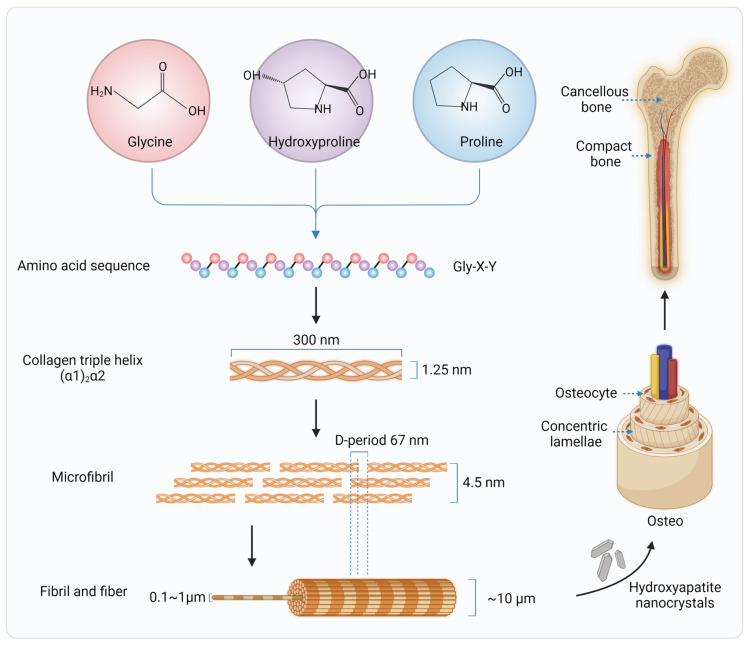
Hierarchical structure of type I collagen fiber and human bone.

**Figure 2 ijms-24-03744-f002:**
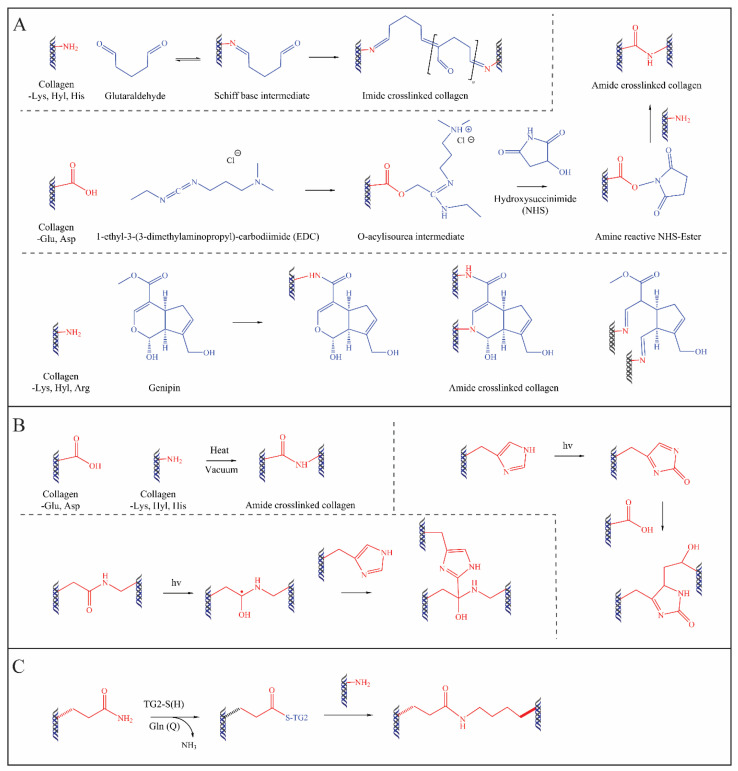
Mechanisms of widely used chemical (**A**), physical (**B**), and biological (**C**) crosslinking technologies of collagen.

**Figure 3 ijms-24-03744-f003:**
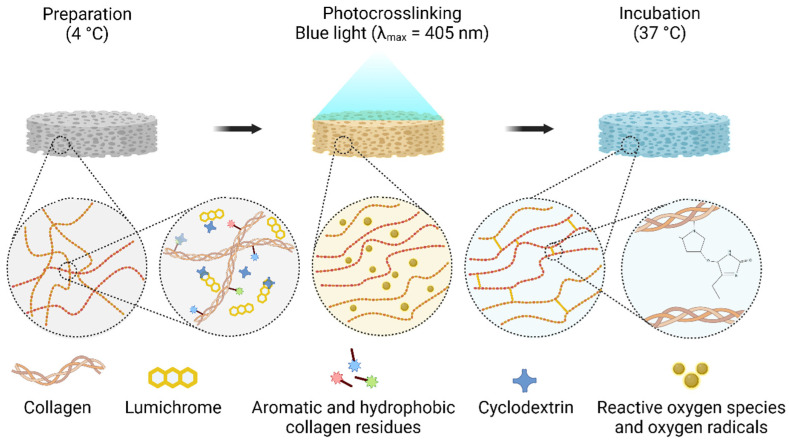
Schematic illustration of LC photo-crosslinking showing the organization and crosslinking of collagen following irradiation (adapted with citation [91]).

**Figure 4 ijms-24-03744-f004:**
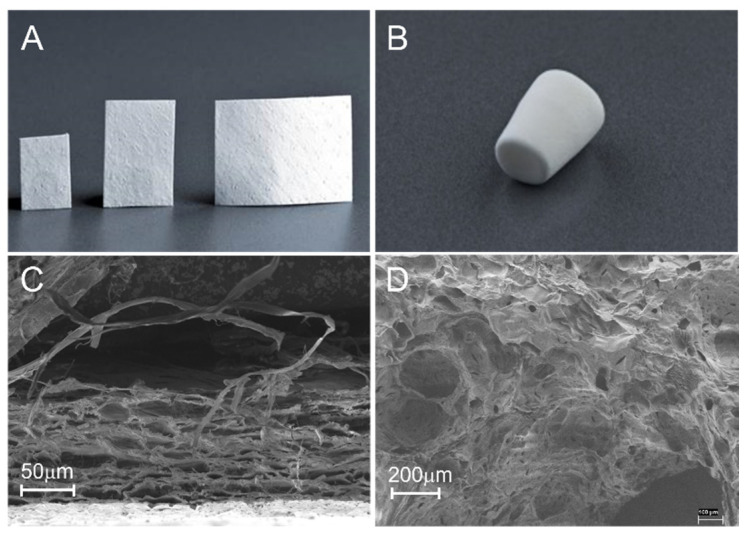
Widely used collagen-based medical devices for bone defect repair. (**A**) Porcine pericardium collagen membrane. (**B**) Lyophilized type I collagen sponge. (**C**) Scanning electron microscope (SEM) imaging of a cross-sectional view of a collagen membrane with a tightly packed smooth side at the bottom and a rough side on the top. (**D**) SEM image of lyophilized porous collagen sponge.

**Figure 5 ijms-24-03744-f005:**
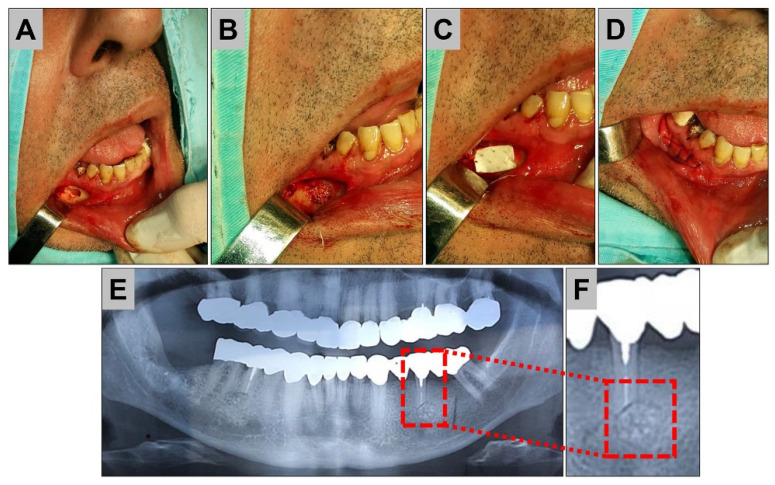
Clinical treatment of bone cysts. (**A**) Exposure of the mandible in the area of the cyst and opening. (**B**) Insertion of a xenogeneic bone substitute material, (**C**) covering with a pericardium-based collagen barrier membrane. (**D**–**F**) Condition after bony healing at 3 months.

**Figure 6 ijms-24-03744-f006:**
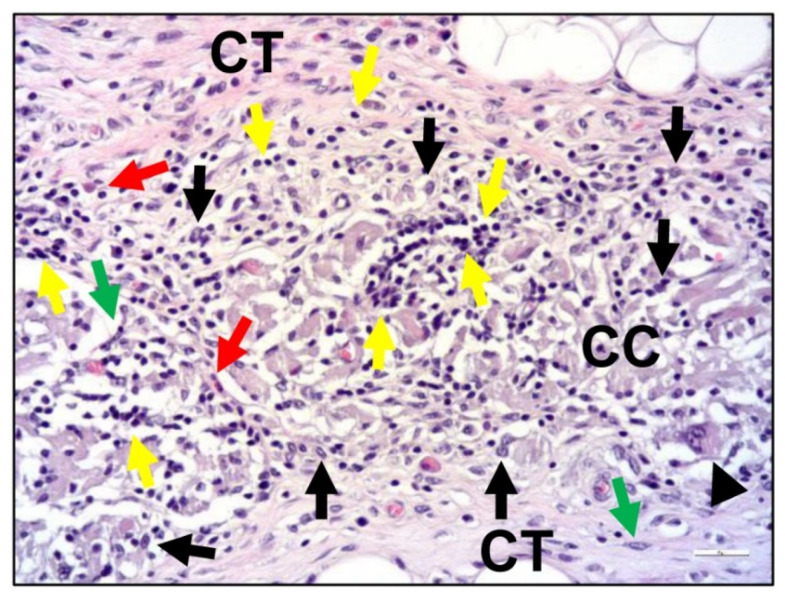
Exemplary histological images of the tissue response to a cross-linked collagen membrane (CC) within the subcutaneous connective tissue (CT) at day 30 post implantation. Black arrows = macrophages, black arrowheads = multinucleated giant cells, green arrows = fibroblasts, red arrows = eosinophilic granulocytes, and yellow arrows = lymphocytes (HE-staining, 200× magnification, scalebar = 20 µm).

**Figure 7 ijms-24-03744-f007:**
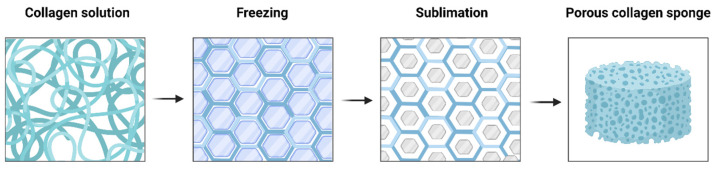
Freeze-drying process of collagen sponge formation. Upon freezing, collagen molecules are entrapped within the developing ice crystals, which have formed into hexagonal structures.

**Figure 8 ijms-24-03744-f008:**
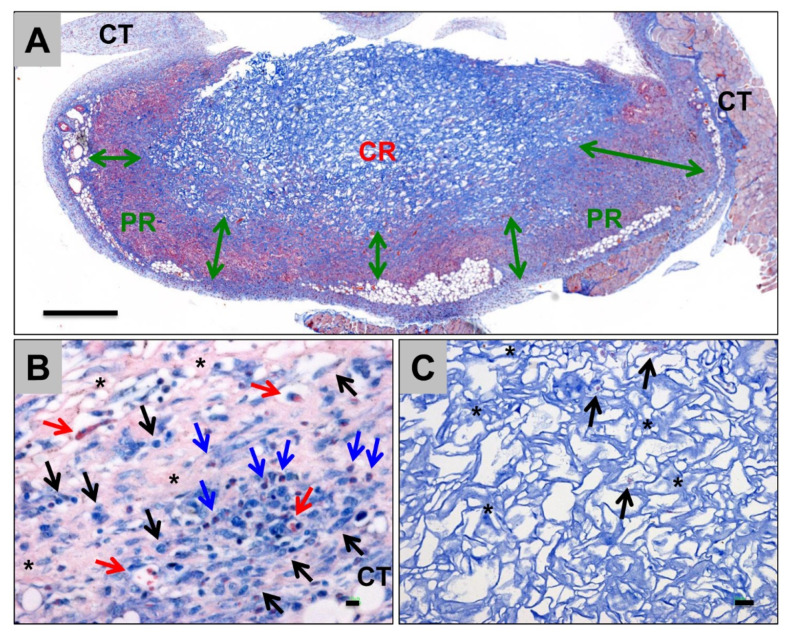
Tissue reactions and integration behavior of a collagen sponge for hemostasis in dental applications at day 15 post implantation within the subcutaneous connective tissue (CT) of Wistar rats. (**A**) Overview of the implantation bed. A peripheral region (PR and double arrows) in which a cellular migration was noticed was separable from a nearly cell-free central region (CR), showing the gradual integration pattern of the biomaterial (Azan-staining, “total scan,” 100× magnification, scalebar = 5mm). (**B**) Tissue reaction within the PR including mainly macrophages (black arrows), as well as lower numbers of eosinophils (blue arrows) in combination with collagen fiber apposition (asterisks) and blood vessel ingrowth (red arrows) (Giemsa-staining, 400× magnification, scalebar = 10 µm). (**C**) Tissue reactions within the CR, including single macrophages (black arrows) within the interspaces of the collagen fibers (asterisks) of the sponge (Azan-staining, 400× magnification, scalebar = 20 µm).

**Figure 9 ijms-24-03744-f009:**
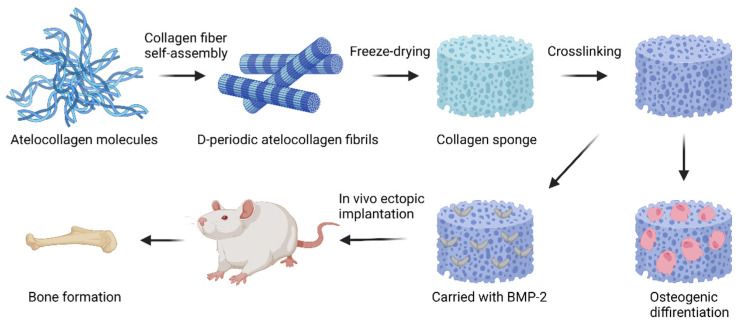
Collagen sponge scaffold built up with self-assembled atelocollagen fibrils, which improved preosteoblast proliferation and differentiation, and bone formation effects in vivo. (Adapted with citation [129]).

**Table 1 ijms-24-03744-t001:** Representative bone implants approved by the FDA for various clinical indications, provided along with their main components and specifications.

Products and Company Name	Main Components	Special Functions and Properties	FDA Code and PMA/510 (k)
Norian skeletal repair system, Synthes USA (West Chester, PA, USA)	Sodium/calcium phosphate	Bone void fillers; non-osteoinduction	MBS; P970010
Ballast MT, SeaSpine Orthopedics Corporation (Carlsbad, CA, USA)	Resorbable Mesh Pouch, e.g., PLGA	Bone void fillers; osteoinduction (w/o human growth factor)	MBP; K200290, K143547 (15 products in total)
Freeman, DePuy Inc. (Raynham, MA, USA)	Polymer/metal/polymer	Semi-constrained cemented prosthesis; knee, patella, femorotibial	MBV; K010212, K884824 (12 products in total)
Insignia Hip Stem, Stryker (Leesburg, VA, USA)	Metal/ceramic/polymer	Semi-constrained cemented or nonporous uncemented prosthesis; hip	MEH; K221104, K220731 (17 products in total)
Subtalar peg implant, Nexa Orthopedics Lundeen subtalar peg implant, Sgarlato Laboratories Inc. (Los Gatos, CA, USA)	Polymer	Metallic bone fixation fastener or plug; subtalar	MJW; K033046, K922292
ANAX 5.5 spinal sys, U&I Corporation (Uijeongbu, Korea)	Metal/polymer, non-porous calcium phosphate	Spondylolisthesis spinal fixation	MNH; K162801, K 162189 (17 products in total)
Sniper Spine Sys, Spine Wave Inc. (Shelton, CT, USA)	Titanium alloy	Spinal pedicle fixation; spinal fixation	MNI; K152174, K152132 (8 products in total)
Ossiofiber, Ossio Ltd. Arthrex Bio, Arthrex Inc. (Naples, FL, USA)	Polymer, e.g., PLDLA	Metallic bone fixation; absorbable	MNU; K212594, K011172
Infuse Bone Graft, Medtronic Sofamor Danek USA, Inc. (Memphis, TN, USA)	Collagen scaffold, recombinant human bone morphogenetic protein 2 (rhBMP-2)	Bone void fillers; osteoinduction	MPW; P000054
ArtFx Medical LLc, NuVasive Incorporated (San Diego, CA, USA)	Titanium alloy Ti6Al-4V ELI	Spinal vertebral body replacement device	MQP; K211892, K202637 (17 products in total)
Altapore MIS, Baxter Healthcare Corporation (Round Lake, IL, USA)	Calcium compounds	Bone void fillers; resorbable calcium salts	MQV; K221644, K213959 (17 products in total)
Infuse Bone Graft, Medtronic Sofamor Danek USA, Inc. (Memphis, TN, USA)	Collagen scaffold, rhBMP-2	Bone void fillers; osteoinduction	NEK; P000058
I-Factor peptide enhanced BG, Cerapedics LLC Augment Injectable, Biomimetic Therapeutics LLC (Franklin, TN, USA)	Synthetic peptides	Bone void fillers	NOX; K140019, K100006
Not available	Calcium compounds	Bone void fillers; resorbable calcium salts	QCG; not available
Infuse Bone Graft and LT-Cage, Medtronic Sofamor Danek USA (Memphis, TN, USA)	Collagen scaffold, metal, rhBMP-2	Bone void fillers; osteoinduction	OJZ; not available
Not available	Calcium compounds; single approved aminoglycoside	Bone void fillers, resorbable; Chronic osteomyelitis of long bones; anti-infection	QRR; not available
Not available	Recombinant platelet-derived growth factor (rhPDGF) and beta-tricalcium phosphate (b-TCP)	Bone void fillers; hindfoot and ankle fusion procedures	QYR; not available
Not available	Bioactive glass particles containing polyethylene glycol and glycerol	Bone void fillers; extremities, posterolateral spine, and pelvis	PBU; not available
Not available	Calcium; synthetic polymers	Bone void fillers; alterable compound for cranioplasty	PJM; not available
Not available	Metals; polyether ether ketone (PEEK)	Bone void fillers; non-alterable compound for cranioplasty, patient-specific preformed implant	PJN; K212414, K190523

**Table 2 ijms-24-03744-t002:** Representative collagen-based bone implants approved by EU, provided along with their main components and specifications.

Products and Company Name	Main Components	Special Function and Properties	Basic UDI-DI/EUDAMED DI
BioCover™, Purgo Biologics Inc. (Seongnam-si, Korea)	Natural fibrous collagen matrix from porcine tendon	Barrier membrane	B-08800039003742, B-08800039003759, B-08800039003766 (10 products in total)
THE Graft™ collagen, Purgo Biologics Inc. (Seongnam-si, Korea)	Porcine-derived bone mineral matrix from cancellous bone and atelocollagen from porcine tendon	Bone block; hydrophilicity, osteoconductivity	B-08800039004992, B-08800039005005, B-08800039005012 (9 products in total)
Striate+ ™, Orthocell Ltd. (Perth, WA, Australia)	Porcine-derived collagen	Barrier membrane; dual layer	B-AU-MF-000018345Z8
ChondroFiller^®^ liquid, Meidrix biomedicals GmbH. (Esslingen am Neckar, Germany)	Rat tail collagen I	Cartilage filler	B-04260349610070, B-04260349610087, B-04260349610179
MaioRegen Prime (Oval)/Slim (Oval)/Chondro+ (Oval), Fin-ceramica Faenza SPA. (Faenza, Italy)	Collagen and hydroxyapatite enriched with magnesium	Bone scaffold; structural biomimetics	B-18054188160963, B-18054188160970, B-18054188160987 (31 products in total)
RegenOss Ortho/Spine, Fin-ceramica Faenza SPA. (Faenza, Italy)	Collagen-hydroxyapatite composite	Bone scaffold; fully biomimetic, highly hydrophilic	B-18054188160529, B-18054188160536, B-18054188160543 (7 products in total)

**Table 3 ijms-24-03744-t003:** The widely applied exogenous chemical, physical, and biological collagen crosslinking methods.

Cross-Linking Methods		Advantages	Disadvantages	Ref.
Chemical	Glutaraldehyde (GA)	Very good mechanical properties and resistance to biodegradation	Significant cytotoxicity and biohazard problems	[51,52,53]
	Hexamethylene diisocyanate	Very good mechanical properties and resistance to biodegradation	Cytotoxicity/inflammation	[54,55]
	1-Ethyl-3-(3-dimethyl aminopropyl)-carbodiimide (EDC) / N-Hydroxysuccinimide (NHS)	Water soluble system, low toxicity/inflammation	Poor biomechanical properties and more rapid biodegradation profiles compared to those of GA-crosslinked examples	[56,57]
	Genipin	Biodegradability and low cytotoxicity/inflammation	Expensive for mass industrial production	[58,59]
	Citric acid	Nontoxic, potential intrinsic mineralization property	Poor biomechanical properties and more rapid biodegradation profiles compared to those of GA-crosslinked examples	[60,61]
Physical	Dehydrothermal treatment	Simple and safe	Denaturation issues; require further modifications	[62,63]
	Plasma treatment	Simple and safe	Denaturation issues; only acceptable for surface modifications	[64,65]
	UV or visible light irradiation	Nontoxic	Denaturation issues	[66,67,68,69]
Biological	Transglutaminase	Nontoxic	Expensive; low stability	[70,71,72]

## Data Availability

All relevant data are included in the manuscript.

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
