# Peer review of "The Use of Collagen-Based Materials in Bone Tissue Engineering"

_ijms, 2023, doi:10.3390/ijms24043744_

Round 1

Reviewer 1 Report

The reviewed manuscript entitled "The Use of Collagen-Based Materials in Bone Tissue Engineering" discuss bone tissue engineering using synthetic bone substitute materials. The authors have described the current status of collagen substitutes approved by the FDA and EU. The article goes on to describe the different types of Collagen and different methods of preparation. The manuscript. The tables in the manuscript are well thought out and smartly compiled. The authors have cleverly used the figure image to illustrate the concepts in a more simplistic way; which makes it more approachable to a wider reader base (Bioengineers, biologists, or clinicians). Best

Author Response

Dear reviewer,

Thank you very much for your professional comments!

Best regards,

Lu Fan

Reviewer 2 Report

In this manuscript, the authors summarized the important role of collagen in native bone tissues, current progress on extraction and modification of collagen, and its applications and challenges in bone tissue regeneration and engineering. This work updates the latest applications of collagen-based materials in bone regeneration and provides some new insights into the potential for further development in the field of bone tissue engineering in the future. However, the authors must clarify the following points before publishing this work in International Journal of Molecular Sciences:

1. For Part 5 (Applications of collagen in bone tissue regeneration and engineering), the authors are suggested to discuss the applications of the collagen-based materials in vitro and in vivo detailedly.

2. Current studies have reported the interactions between the RGD and human mesenchymal stem cells, as well as modulating the adhesion, proliferation, and differentiation of stem cells by tuning the tether mobility (Nano Letters 17 (3), 1685-1695), anisotropic nanoscale presentation of RGD (Advanced Functional Materials 29 (8), 1806822), and stiffness of hydrogel (J. Compos. Sci. 2022, 6(1), 19). The authors should add additional discussions on these fundings and cite these papers accordingly.

3. There is a typo “o” on Page 13, Line 399.

Author Response

Dear reviewer,

Thank you very much for your professional advice and suggestion! Based on your comments, we have carefully revised our manuscript.

  1. The applications of collagen-based materials in vitro and in vivo were discussed more in detail in Page 9, Line 270-278.
  2. The advanced studies about the interactions between RGD of bone regeneration related cells and biomaterials have been further discussed and cited in Page 13, Line 373-376 and Page 14, Line 416-419.
  3. We also checked and revised the writing and grammar of the full manuscript (e.g., Page 13, Line 399). We added and adjusted references as well.

Best regards,

Lu Fan